# Feature selection based on neighborhood rough sets and Gini index

Yuchao Zhang, Bin Nie, Jianqiang Du, Jiandong Chen, Yuwen Du, Haike Jin, Xuepeng Zheng, Xingxin Chen and Zhen Miao

School of Computer Science, Jiangxi University of Chinese Medicine, NanChang, JiangXi, China



## ABSTRACT

Neighborhood rough set is considered an essential approach for dealing with incomplete data and inexact knowledge representation, and it has been widely applied in feature selection. The Gini index is an indicator used to evaluate the impurity of a dataset and is also commonly employed to measure the importance of features in feature selection. This article proposes a novel feature selection methodology based on these two concepts. In this methodology, we present the neighborhood Gini index and the neighborhood class Gini index and then extensively discuss their properties and relationships with attributes. Subsequently, two forward greedy feature selection algorithms are developed using these two metrics as a foundation. Finally, to comprehensively evaluate the performance of the algorithm proposed in this article, comparative experiments were conducted on 16 UCI datasets from various domains, including industry, food, medicine, and pharmacology, against four classical neighborhood rough set-based feature selection algorithms. The experimental results indicate that the proposed algorithm improves the average classification accuracy on the 16 datasets by over 6%, with improvements exceeding 10% in five. Furthermore, statistical tests reveal no significant differences between the proposed algorithm and the four classical neighborhood rough set-based feature selection algorithms. However, the proposed algorithm demonstrates high stability, eliminating most redundant or irrelevant features effectively while enhancing classification accuracy. In summary, the algorithm proposed in this article outperforms classical neighborhood rough set-based feature selection algorithms.

Corresponding author
Bin Nie, ncunb@163.com

## INTRODUCTION

In data mining and machine learning, the goal of feature selection is to choose the most representative and valuable features from the original dataset to improve the performance and interpretability of models. In classification problems, a crucial feature selection step is establishing an effective feature evaluation function for candidate feature subsets. Common feature evaluation functions currently include consistency (*Shin & Miyazaki, 2016*), correlation (*Gao et al., 2018*; *Malhotra & Jain, 2022*), information gain (*Zhang et al., 2020*; *Prasetiyowati, Maulidevi & Surendro, 2022*; *Shu et al., 2023*), mutual information (*Gao, Hu & Zhang, 2020*; *Lall et al., 2021*), classifier error rate (*Got, Moussaoui & Zouache, 2021*; *Moslehi & Haeri, 2020*; *Solorio-Fernández, Carrasco-Ochoa & Martínez-Trinidad,*

*2016*; *Li et al., 2021*), distance (*Lee & Oh, 2016*), Gini index (*Park & Kwon, 2011*; *Manek et al., 2017*; *Liu, Zhou & Liu, 2019*), *etc.*

Rough set theory, proposed by *Pawlak (1982)* and continuously improved by subsequent researchers, is a mathematical tool for dealing with uncertain and inexact information. Recently, this theory has been widely applied to feature selection in data mining and machine learning (*Sang et al., 2022*; *Huang, Li & Qian, 2022*; *Zhang et al., 2023*; *Wan et al., 2023*; *Yang et al., 2023*). The rough set theory divides the dataset into equivalence classes to reveal the dependency relationships among attributes and the process of generating decision rules. For classification problems, rough set theory uses features to induce binary relations and divides samples into different information granules based on these binary relations. These information granules are then used to approximate decision variables and represent upper and lower approximations of decisions. Based on this, a feature evaluation function called the dependency function is defined. Different types of binary relations lead to different granulation mechanisms, resulting in various rough set models, such as classical rough set (*Pawlak, 1982*), similarity relation-based rough set (*Dai, Gao & Zheng, 2018*), dominance relation-based rough set (*Greco, Matarazzo & Slowinski, 1999*; *Shao & Zhang, 2004*), fuzzy rough set (*Pawlak, 1985*), and other rough set models (*Sang et al., 2018*).

The neighborhood rough set (NRS) (*Hu et al., 2008*) is one of the most crucial rough set models proposed to address the challenges of handling continuous features in classical rough set theory. Since its application to feature selection (*Hu et al., 2008*), NRS has gained widespread attention in data mining and machine learning (*Liu et al., 2016*; *Zeng, She & Niu, 2014*). Many scholars have proposed different feature evaluation functions based on this model and developed corresponding feature selection algorithms. *Hu et al. (2011)* proposed neighborhood information entropy to address the fact that Shannon entropy cannot directly evaluate uncertainty on continuous features. *Wang et al. (2018)* explored some neighborhood distinguishability measures to assess data uncertainty. They proposed the K-nearest neighbor NRS by combining the advantages of neighborhood and K-nearest neighbors while focusing on data distribution (*Wang et al., 2019b*). *Wang et al. (2019a)* proposed neighborhood self-information to utilize deterministic and uncertain information better. *Sun et al. (2019a, 2019b)* introduced the Lebesgue measure into NRS, enabling feature selection on infinite sets (*Sun et al., 2019a*) and incomplete sets (*Sun et al., 2019b*). *Li, Xiao & Wang (2018)* extended discernibility matrices to NRS and applied them to a power system stability assessment.

Many of the evaluation methods above extend the evaluation metrics for discrete features to continuous random variables through neighborhood relations. For example, neighborhood information entropy extends Shannon entropy to continuous random variables, neighborhood discernibility matrix extends rough set discernibility matrix, and neighborhood self-information extends self-information to continuous random variables. The Gini index (GI) (*Breiman et al., 1984*), first introduced by Breiman in 1984 and applied to node splitting in decision trees, accurately quantifies the impurity of a dataset. Especially in classification problems, it effectively measures the contribution of features to

classification results and has been widely used in feature selection in data mining and machine learning (*Breiman, 2001*; *Wang, Deng & Xu, 2023*).

This article combines NRS with GI from two perspectives and proposes two unique feature importance evaluation metrics. First, from the standpoint of sample neighborhoods, the Neighborhood Gini index is proposed to measure the importance of features through neighborhood information. Second, from the standpoint of class neighborhoods, the Neighborhood Class Gini index is proposed to reveal the differences in features among different classes. The properties of these two evaluation metrics and their relationships with attributes are discussed. Based on these evaluation metrics, two forward greedy algorithms are designed for feature selection. Finally, the effectiveness and stability of the proposed algorithms are validated through experiments.

The structure of this article is as follows: In the "Materials and Methods" section, a review of the fundamental concepts of NRS and GI is provided, and the combination of NRS and GI is used to propose two distinct feature importance evaluation indicators. The properties of these two evaluation indicators and their relationships with attributes are discussed. Subsequently, the importance of candidate features is defined based on the two evaluation indicators. Building upon this, two separate forward greedy feature selection algorithms are formulated. In the "Experimental Analysis and Discussion" section, the effectiveness and stability of the proposed algorithms are verified. In the "Conclusions" section, we concluded the article with possible directions for future research.

## MATERIALS AND METHODS

### Neighborhood rough set

In rough sets, information tables are often represented by $<U, A>$, where $U = \{x_1, x_2, ..., x_n\}$ is a non-empty finite set of samples and $A = \{a_1, a_2, ..., a_m\}$ is a non-empty finite set of attributes used to describe those samples.

Let $<U, A>$ be a table of information, $B \subseteq A$ and $d_B$ is a binary functional relation defined on $U$ with an attribute subset B, that is, $d_B : U \times U \rightarrow R^+$. Then, $d_B$ is said to be a distance metric on $U$ when it satisfies the following relation:

(1) $d_B(x_1, x_2) \geq 0$, $d_B = 0$ if and only if $x_1 = x_2$, $\forall x_1, x_2 \in U$;
(2) $d_B(x_1, x_2) = d_B(x_2, x_1)$, $\forall x_1, x_2 \in U$;
(3) $d_B(x_1, x_3) \leq d_B(x_1, x_2) + d_B(x_2, x_3)$, $\forall x_1, x_2, x_3 \in U$.

The Euclidean distance is a commonly used distance measure, and all the subsequent distance references in this article are in terms of the Euclidean distance. For any two samples, the calculation of the Euclidean distance is as follows:

$$d_B(x_i, x_j) = \sum_{a \in B} \sqrt{(x_i^a - x_j^a)^2} \tag{1}$$

In an information table $<U, A>$, for any sample $x \in U$, attribute subset $B \subseteq A$, the neighborhood similarity relation $R_B^\sigma$ is defined as follows:

$$R_B^\sigma = \{(x, y) \in U \times U | d_B(x, y) \leq \sigma\} \tag{2}$$

where $\sigma \geq 0$ is a user-defined constant. For any $x \in U$, its neighborhood similarity class $[x]_B^\sigma$ is defined as follows:

$$[x]_B^\sigma = \{y \in U : (x, y) \in R_B^\sigma\} \tag{3}$$

In an information system, neighborhood similarity classes are also referred to as neighborhood information granules, abbreviated as neighborhood granules. Here, $\sigma$ is called the radius of the neighborhood granules. In the information table $<U, A>$, the neighborhood granule family $\{[x_i]_B^\sigma | i = 1, 2, ..., n\}$ forms a covering of $U$. The neighborhoods of all objects in the domain constitute the granulation of the domain, and the neighborhood granule family constitutes the fundamental concept system in the domain space. Through these fundamental concepts, we can approximate any concept in the space.

For any sample set $X \subset U$, its lower approximation $\underline{R}_B^\sigma$ and upper approximation $\bar{R}_B^\sigma$ are defined as follows:

$$\underline{R}_B^\sigma = \{x \in U : [x]_B^\sigma \subseteq X\} \tag{4}$$
$$\bar{R}_B^\sigma = \{x \in U : [x]_B^\sigma \cap X \neq \varnothing\} \tag{5}$$

Let $D$ be a classification decision attribute defined on $U$, and $A \cap D = \varnothing$. In this case, the triple $<U, A, D>$ is referred to as a decision table. In the decision table $<U, A, D>$, attribute D divides U into $r$ decision classes, denoted as $U/D = \{E_1, E_2, ..., E_r\}$. Here, $E_i(i = 1, 2, ..., r)$ is called a general equivalence class, meaning that all the samples in $E_i$ have the same class labels. In a decision table $<U, A, D>$, where $B \subseteq A$, and $R_B^\sigma$ is a neighborhood similarity relation defined on the attribute set $B$ in $U$ with a neighborhood radius of $\sigma$, the upper approximation $\bar{R}_B^\sigma(D)$ and lower approximation $\underline{R}_B^\sigma(D)$ of the decision attribute $D$ with respect to the attribute set $B$ at a neighborhood granule size of $\sigma$ are defined as follows:

$$\bar{R}_B^\sigma(D) = \cup_{k=1}^r \bar{R}_B^\sigma(E_k) \tag{6}$$
$$\underline{R}_B^\sigma(D) = \cup_{k=1}^r \underline{R}_B^\sigma(E_k) \tag{7}$$

The positive domain of the decision table is written as:

$$POS_B^\sigma(D) = \cup_{E_k \in U/D} \underline{R}_B^\sigma(E_k) \tag{8}$$

The boundary domain of the decision table is written as:

$$Rn_B^\sigma(D) = U_B(D) - POS_B^\sigma(D) \tag{9}$$

The dependency function $\gamma_B^\sigma(D)$ of D associated with B is formulated as:

$$\gamma_B^\sigma(D) = \frac{|POS_B^\sigma(D)|}{|U|} \tag{10}$$

where $|.|$ indicates the cardinality of a set.

## Gini index

GI is a metric used to measure the impurity of a dataset and is commonly employed in feature selection for decision tree algorithms. The values of GI range from 0 to 1. When GI = 0, the dataset's impurity is minimal, meaning all elements in the dataset are the same. Conversely, when GI = 1, the dataset's impurity is maximal, indicating that all elements in the dataset are different. For a dataset $D$ with $r$ categories, where each category's proportion of samples is denoted as $p_i$, the formula for calculating GI is as follows:

$$GI(D) = 1 - \sum_{i=1}^{r} p_i^2 \tag{11}$$

GI evaluates the impurity of a dataset based on the distribution of class probabilities to determine the importance of the corresponding features. A smaller GI indicates higher dataset purity and better discriminative power of the feature. However, in the same dataset, different feature subsets result in the same class probability distribution, making it unsuitable for directly evaluating the classification performance of different feature subsets. Therefore, a new influencing factor must be introduced to make the class probability distributions vary across feature subsets. For instance, in the Classification and Regression Trees, a tree-like structure is introduced to partition the dataset. This partitioning leads to substantial differences between data subsets created by different feature divisions, resulting in distinct class probability distributions. This makes the Gini index enable the measurement of feature importance. In NRS, when the neighborhood radius is consistent, different attribute sets lead to distinct neighborhoods for samples. Conversely, when the attribute set is constant, various neighborhood radii correspond to different neighborhoods. Different neighborhoods could lead to varying class probability distributions. GI values would also differ. This makes GI applicable for feature selection in NRS.

Next, two different feature importance evaluation metrics integrating NRS and GI will be proposed.

## The proposed method

### Neighborhood Gini index

Samples with certain similarities should be grouped into the same category, and samples within the neighborhood range of a sample are considered similar from a distance perspective. Their features determine the similarity of samples. However, for reasons such as data collection, some features may be redundant or irrelevant to class labels. Therefore, the class labels of samples within a neighborhood range may not be consistent under a subset of features. It is necessary to select features that can effectively represent the characteristics of all categories so that the class labels within the neighborhood range of samples are as consistent as possible.

The more consistent the class labels of samples within the neighborhood range, or the higher the purity of classes within the neighborhood range, the better the corresponding subset of features can represent that class. At this point, the subset of features can represent

the local characteristics of that class well. If a subset of features can represent all the local characteristics of all classes well, *i.e.*, the class purity within the neighborhood range of all samples in the dataset is high, then the subset of features can distinguish all classes well. In this case, the importance of features is also higher.

Based on this idea, we use GI to represent the impurity of the dataset and propose the Neighborhood Gini index (NGI). NGI evaluates the importance of a subset of features by assessing the purity of all samples' neighborhood ranges under that feature subset. The definition of NGI is given below:

**Definition 1:** Given a decision table $< U, A, D >$, for any $x_i \in U, B \subseteq A$, the impurity of $\sigma$ neighborhood $R_B^\sigma(x_i)$ is defined as:

$$GI_B^\sigma(x_i) = 1 - \sum_{j=1}^{r} p_j^2 \tag{12}$$

where $r$ represents the number of categories, and $p_j$ signifies the proportion of the jth category within the $\sigma$ neighborhood of $x_i$.

Throughout the decision table, the impurity of the decision table is the mean value of the impurity within the neighborhood of each sample:

$$NGI_B^\sigma(D) = \frac{1}{n} \sum_{i=1}^{n} GI_B^\sigma(x_i) \tag{13}$$

From Definition 1, it can be observed that NGI is influenced by two parameters: the feature subset $B$ and the neighborhood radius $\sigma$. As GI focuses on the distribution of classes, changes in the number of samples within the neighborhood range can lead to class distribution changes, thereby affecting GI's magnitude. However, the behavior of GI to changes in the feature subset and neighborhood radius is not strictly monotonic. The following will analyze the variations of NGI to changes in the feature subset and neighborhood radius separately.

*Impact of feature on NGI*

For any subset of features $B_1 \subset B_2 \subseteq A$, adding one or more features to $B_1$ to obtain $B_2$ does not necessarily guarantee that $NGI_{B_2}(D)$ will always be smaller than $NGI_{B_1}(D)$, and the process of its change is not completely monotonous, as shown in Fig. 1.

Figure 1 shows the relationship between different sizes of feature subsets and NGI under the same neighborhood. The x-axis is the number of features, and the y-axis is the NGI of the corresponding feature. The smaller feature subset is a proper subset of the larger feature subset. In Fig. 1A, the feature subset consists of 18 continuous features, namely [17, 59, 1, 18, 51, 30, 21, 10, 52, 20, 50, 39, 53, 55, 29, 54, 48, 25], and the data is the "Sonar" dataset from the UCI Machine Learning Repository (*Kelly, Longjohn & Nottingham, 1998*). The neighborhood radius in Fig. 1A is set to 0.15. In Fig. 1B, the feature subset comprises 18 discrete features, namely [4, 7, 15, 10, 9, 13, 14, 6, 17, 16, 0, 12, 3, 2, 5, 8, 11, 1], and the data is the "anneal" dataset from the UCI Machine Learning Repository. The neighborhood radius in Fig. 1B is set to 0.4.
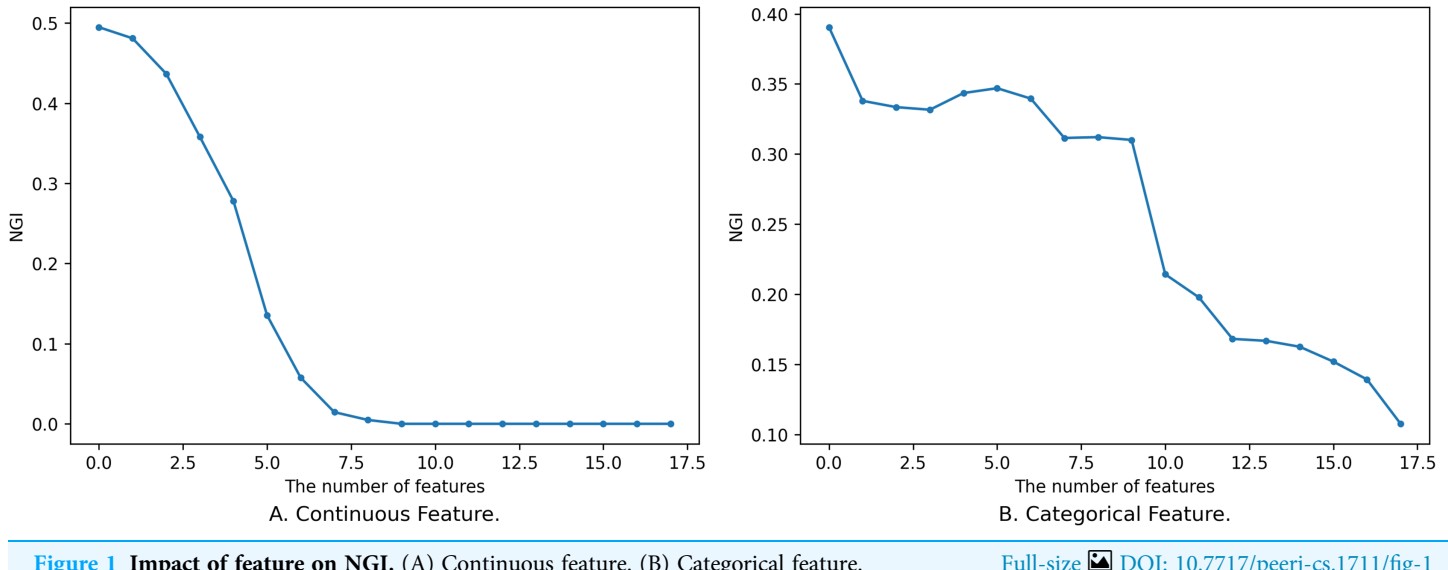

**Figure 1** **Impact of feature on NGI.** (A) Continuous feature. (B) Categorical feature.

When the features are continuous, the variation in the samples within the neighborhood range is small, leading to minor changes in the class distribution. As a result, the variation curve is relatively smooth, as shown in Fig. 1A. However, when the features are discrete, introducing new features might drastically reduce the number of samples within the neighborhood range, causing larger changes in the class distribution. This results in a fluctuating variation curve, as depicted in Fig. 1B.

From Fig. 1, it is evident that the variation of NGI is not strictly monotonic at a local level, yet it generally exhibits a descending trend as a whole. This phenomenon can be attributed to the overarching effect that, with an increase in the number of features, the data provides a more precise portrayal of the samples, making their inherent characteristics more prominent. When the features a sample emphasizes are more aligned with its class attributes, the purity of the sample's neighborhood increases, leading to a smaller NGI. Conversely, when the emphasized features deviate from the class attributes, the neighborhood's purity decreases, resulting in a larger NGI. As the number of features expands, characteristics relevant to the class gradually come into sharper focus, consequently contributing to the observed overall decreasing trend.

It is worth noting that not all continuous feature subsets follow smooth and monotonic variation curves, and not all discrete features yield fluctuating curves. Continuous features might also exhibit fluctuations, while discrete features can exhibit smooth and monotonic behaviors. However, regardless of whether the features are continuous or discrete, the overall tendency is characterized by a decrease.

The subsequent explanation illustrates the variation of NGI through changes in the class distribution within the feature space neighborhood of sample $x_i$:

We utilize the change in GI within the neighborhood feature subspace of sample $x_i$ to symbolize the overall changes in NGI across the entire dataset. The distribution of samples

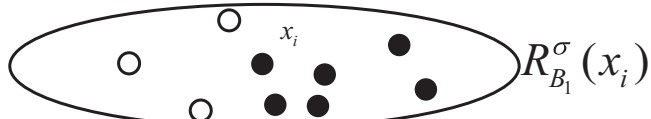

**Figure 2** Sample distribution in original neighborhood feature subspace.

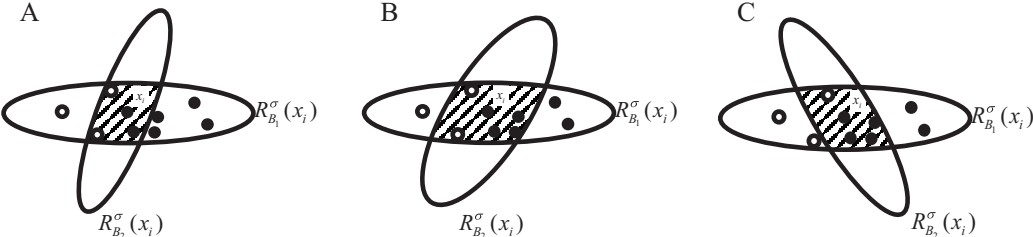

**Figure 3** (A–C) Sample distribution in the neighborhood feature subspace after adding one feature.

within this localized neighborhood feature subspace is depicted in Fig. 2. Among them, the hollow circle class accounts for $\frac{1}{3}$, and the solid circle class accounts for $\frac{2}{3}$. At this time, $NGI_B^\sigma(x_i) = 1 - \frac{1^2}{3} - \frac{2^2}{3} = \frac{4}{9}$. Let $a_i \in A - B$, $NGI_{B \cup \{a_i\}}^\sigma$ change relative to $NGI_B^\sigma$ as follows:

1. NGI increases when the number of samples in the neighborhood feature subspace with a large proportion of categories decreases proportionally more than the number of samples with a small proportion of categories. As shown in Fig. 3A, the number of samples in the hollow circle category decreases by 1, at which point the percentage is $\frac{2}{4}$, and the number of samples in the solid circle category decreases by 4, at which point the percentage is $\frac{2}{4}$, $NGI_B^\sigma = \frac{4}{9} < NGI_{B \cup \{a_1\}}^\sigma = 1 - \frac{2^2}{4} - \frac{2}{4} = \frac{1}{2}$;

2. NGI remains unchanged when there is no change in the samples in the neighborhood feature subspace or when the samples in the neighborhood, according to the proportion of categories in equal, are reduced. As shown in Fig. 3B, at this time the proportion of the hollow circle category is still $\frac{1}{3}$, and the proportion of the solid circle category is $\frac{2}{3}$, $NGI_{B \cup \{a_2\}}^\sigma = NGI_B^\sigma = \frac{4}{9}$;

3. NGI decreases when the number of samples in the neighborhood feature subspace with a large proportion of categories decreases proportionally less than the number of samples with a small proportion of categories. As shown in Fig. 3C, the hollow circle category samples are reduced by 2, at this time the proportion of $\frac{1}{5}$, the solid circle category samples are reduced by 2, at this time the proportion of $\frac{4}{5}$, $NGI_B^\sigma = \frac{4}{9} > NGI_{B \cup \{a_3\}}^\sigma = 1 - \frac{1^2}{5} - \frac{4^2}{5} = \frac{8}{25}$.

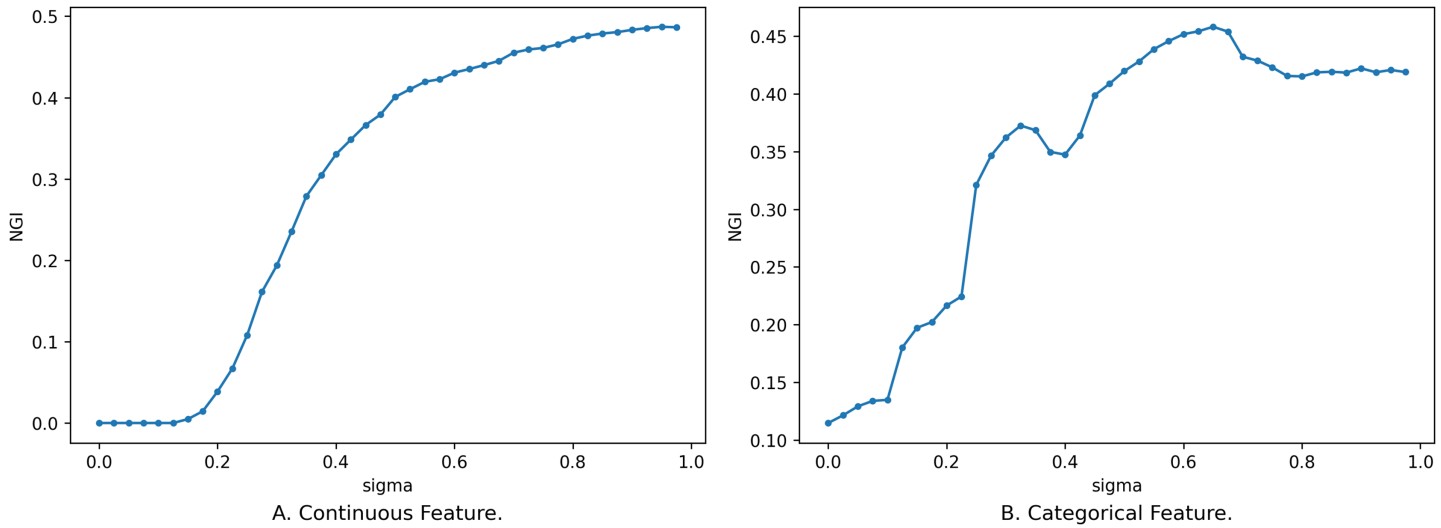

**Figure 4 Impact of neighborhood radius on NGI.** (A) Continuous feature. (B) Categorical feature.

### Impact of neighborhood radius on NGI

In addition to the influence of feature subsets on NGI, the size of the neighborhood radius also affects the changes in the distribution of classes within the neighborhood feature subspace, consequently impacting the magnitude of NGI. So, we have delved into the impact of varying neighborhood radius sizes on NGI. We set the neighborhood radius to range from 0 to 1 with a step size of 0.025, and the relationship between the neighborhood radius and NGI is depicted in Fig. 4.

Figure 4 illustrates the relationship between different sizes of the neighborhood radius and NGI for the same feature subset. The x-axis is the size of the neighborhood radius, and the y-axis is the NGI of the corresponding neighborhood radius. In Fig. 4A, the feature subset consists of 10 continuous features, namely [46, 8, 3, 2, 44, 53, 59, 24, 25, 42], sourced from the "Sonar" dataset in the UCI Machine Learning Repository. In Fig. 4B, the feature subset comprises five discrete features, namely [7, 14, 10, 1, 17], sourced from the "anneal" dataset in the UCI Machine Learning Repository.

As the value of $\sigma$ gradually increases, the number of samples within the neighborhood range also increases, leading to a rise in impurity. When $\sigma$ is relatively small, the change in the number of samples within the neighborhood is small, and the newly added samples are mostly from the same category. Consequently, the change curve remains relatively stable. When $\sigma$ exceeds a certain threshold (as depicted in Fig. 4A, *e.g.,* 0.195), the category labels of the newly added samples start to deviate from those of the original samples. This leads to a change in NGI, eventually converging to the GI of the entire dataset. While an overall trend increases as the neighborhood radius gradually enlarges, this change is not necessarily monotonic. The reasons behind the variation of NGI with $\sigma$ are analogous to the reasons for its variation with the size of the feature subset. These reasons will not be reiterated here.

### Neighborhood Class Gini index

In the context of neighborhood rough sets based on decision tables, the upper approximation of a category refers to the set of samples within the neighborhood range of that category. This sample set includes all samples from the current category and some from others. It is obvious that the fewer categories in the upper approximation and the fewer samples from other categories, the higher the purity of the upper approximation of the category. A higher purity indicates that the corresponding feature provides a more accurate description of that category, making it easier to distinguish it from others. If the upper approximations of all categories have higher purity, all types within the dataset can be better distinguished, and the corresponding features are more important. Based on this principle, this article proposes the Neighborhood Class Gini index (NCGI). It evaluates features' importance by assessing the upper approximation's impurity under different feature subsets. The definition of NCGI is provided below:

**Definition 2:** Given a decision table $< U, A, D >$, let $B \subseteq A$, $E_k \in U/D(k = 1, 2, ..., r)$, and $\bar{R}_B^\sigma(E_k)$ is the upper approximations of $E_K$, so the impurity of $\bar{R}_B^\sigma(E_k)$ is defined as:

$$GI_B^\sigma(\bar{R}_B^\sigma(E_k)) = 1 - \sum_{i=1}^{r} p_i^2 \tag{14}$$

In the entire decision table, the impurity of the decision table is the average of the impurities of all category upper approximations:

$$NCGI_B^\sigma(D) = \frac{1}{r} \sum_{k=1}^{r} GI_B^\sigma(\bar{R}_B^\sigma(E_k)) \tag{15}$$

Similar to NGI, the magnitude of NCGI is also influenced by the neighborhood radius $\sigma$ and the feature subset $B$. The following comparison illustrates the changes in the two evaluation metrics concerning the number of features and the neighborhood radius. The data in Fig. 5 corresponds to the data in Fig. 1, while the data in Fig. 6 corresponds to that in Fig. 4.

In the case of continuous features, the trend of NCGI with changing $\sigma$ is closely similar to that of NGI, displaying relatively smooth changes. NCGI exhibits a lower overall sensitivity to variations in the number of features and the neighborhood radius yet displays higher sensitivity within certain intervals, such as when $\sigma$ ranges from 0.2 to 0.375 in Fig. 5A. This phenomenon stems from NGI being rooted in the sample neighborhood, with the class distribution altering as the neighborhood radius expands. Conversely, NCGI assesses feature importance from a class neighborhood perspective. As the neighborhood radius expands, the number of samples within the neighborhood increases. However, when $\sigma$ is small, the newly added samples within the neighborhood share the same category as the current sample. Consequently, the category distribution in the upper approximation remains unchanged. When $\sigma$ is big enough, the upper approximation of the class encompasses all samples within the dataset, resulting in NCGI equating to the overall GI and ceasing to change with variations in the neighborhood radius.

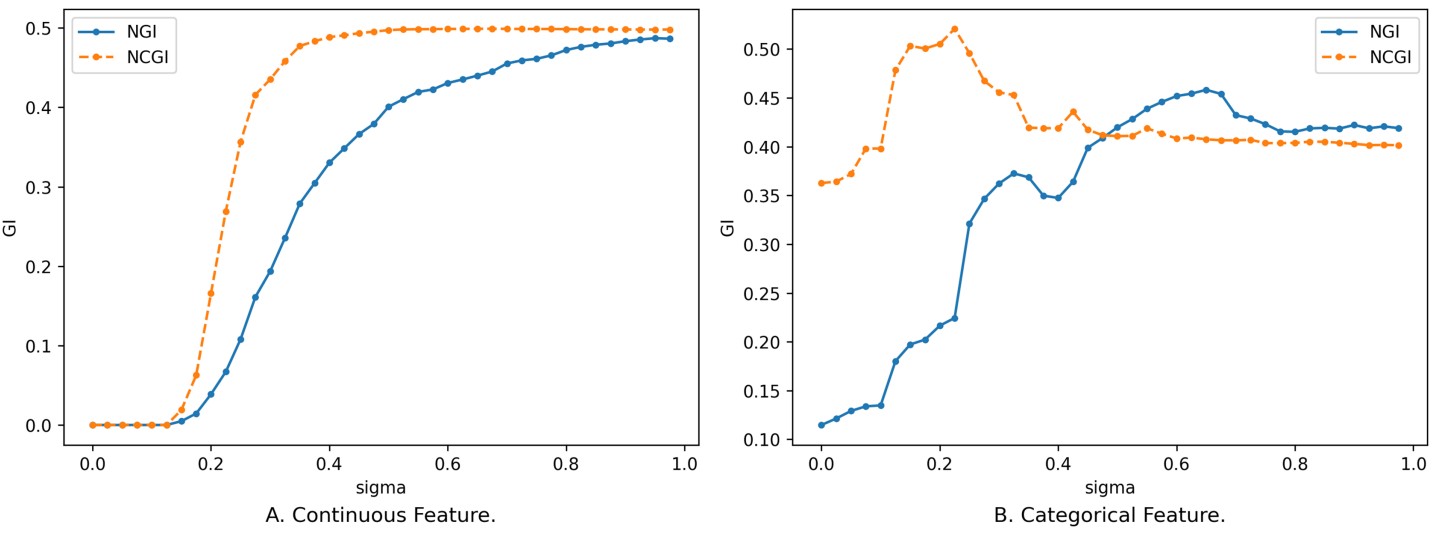

**Figure 5 Impact of neighborhood radius on two evaluation metrics.** (A) Continuous feature. (B) Categorical feature.

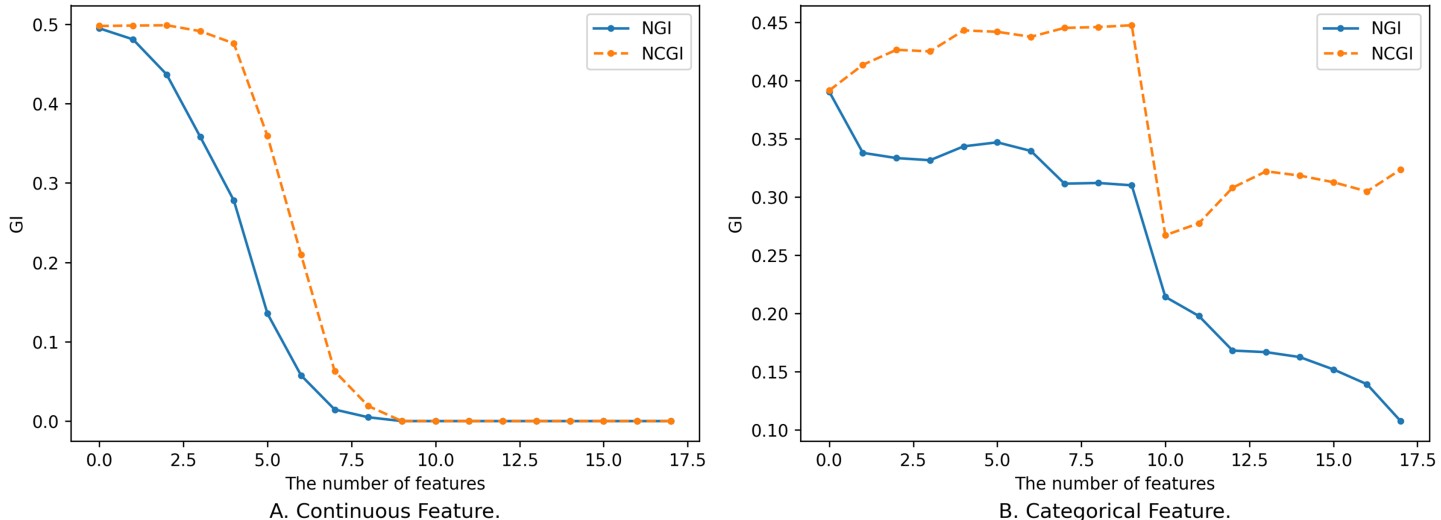

**Figure 6 Impact of feature on two evaluation metrics.** (A) Continuous feature. (B) Categorical feature.

In the case of discrete data, as the neighborhood radius varies, a sudden influx of samples within the neighborhood range can significantly alter the class distribution, causing larger fluctuations in the change curve, particularly noticeable in Fig. 5B. However, overall, NCGI experiences smaller changes in amplitude compared to NGI.

The number of samples within the neighborhood range gradually decreases with increased features. In the context of continuous features, the reduction in sample count is relatively smooth, as shown in Fig. 6A. Consequently, the class distribution alteration of neighborhoods is similarly gradual. With the increase of purity within the neighborhood, GI decreases until it converges to 0.

For categorical features, the introduction of new features can exert a substantial influence on the class distribution within the neighborhood, leading to larger fluctuations, as shown in Fig. 6B. This is particularly evident upon the inclusion of the 11th feature, where both NGI and NCGI exhibit a sharp decline. This decline implies that adding this feature enhances the purity within the neighborhood, facilitating the differentiation of various categories. Beyond the 11th feature, sample neighborhoods and class neighborhoods' results diverge. These features can decrease the impurity within the sample neighborhood but paradoxically lead to an increase in the impurity within the class neighborhood.

### Feature selection

**Definition 3:** Given a decision table $<U, A, D>$, $B \subseteq A$, $a_i \in A - B$, the importance of $a_i$ with respect to $B$ is calculated as follow:

$$SIG(a_i, B, D) = GI_B^\sigma(D) - GI_{B \cup \{a_i\}}^\sigma(D) \tag{16}$$

where $GI_B^\sigma$ stands for NGI and NCGI proposed in this article.

In Definition 3, we defined the importance of feature $a_i$ relative to a given feature subset B. In the case where feature subset B is known, it is adding a feature $a_i$ to B and observing its GI (which refers to either NGI or NCGI). If GI decreases, it indicates that $a_i$ is a crucial feature relative to B. Conversely, if the GI remains unchanged or increases, it suggests that $a_i$ is a redundant feature relative to B or even an irrelevant feature with respect to the decision table $<U, A, D>$.

To achieve better classification performance, we aim to select each feature $a_i$ in such a way that it is the most crucial feature relative to B. Therefore, we have designed heuristic algorithm based on Neighborhood Gini index (HANGI) and heuristic algorithm based on Neighborhood Gini index (HANCGI) feature selection algorithms using a forward greedy approach to select the optimal feature subset. The two algorithms differ only in calculating $SIG(a_i, B, D)$, and their processes are illustrated in Fig. 7.

In HANGI and HANCGI, the algorithm starts by taking as input a decision table $<U, A, D>$, a neighborhood radius $\sigma$, and a minimum threshold $\beta$ for the relative importance of candidate features with respect to the reduced subset. Subsequently, the reduced subset and the candidate feature subset are initialized. An evaluation is made to determine if the candidate feature subset is empty. The current reduced subset is directly output if the candidate feature subset is empty. Conversely, if the candidate feature subset is not empty, all candidate features are iterated through. Each candidate feature's importance concerning the reduced subset is computed using NGI or NCGI, denoted as $SIG(a_i, red, D)$. The feature with the highest importance is selected, and its importance is marked as $SIG_{max}$. Following this, an assessment is carried out to determine whether $SIG_{max} > \beta$. If true, the feature with the highest importance is removed from the candidate feature subset and incorporated into the reduced subset. The candidate feature subset is then revisited. Output the reduced subset until the candidate feature subset is empty or $SIG_{max} \leq \beta$.

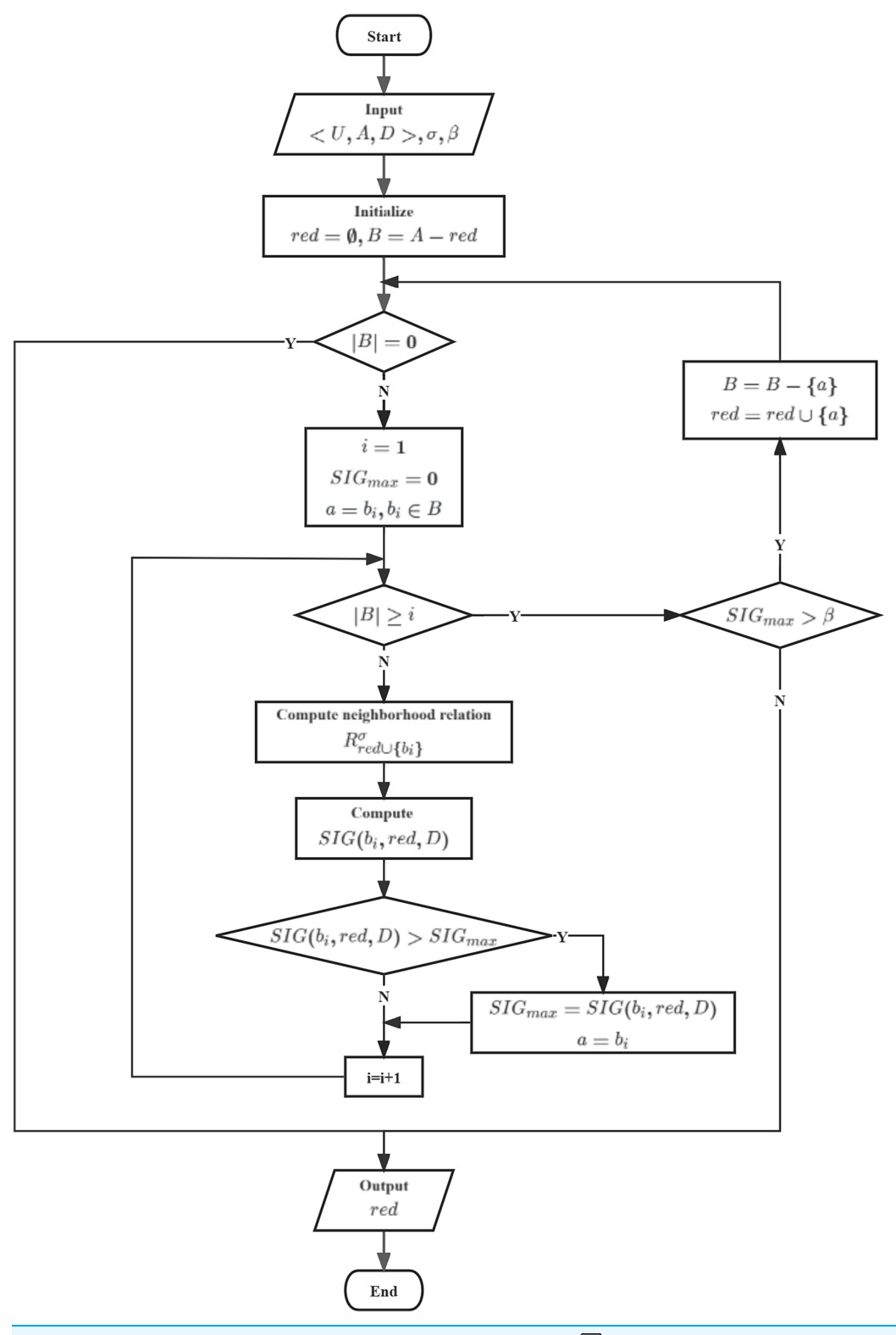

**Figure 7 Flowchart of HANGI and HANCGI.**

Assuming a dataset contains n samples, m features, and r categories, the best feature in each iteration is the one with the longest search time, with a worst-case search time of $(m^2 + m)/2$. Calculate the time $n(n-1)/2$ required to determine the neighborhood relationship between samples in the dataset. The time to compute the Gini index within the neighborhood range is also nr. Therefore, the time complexity of the NGI and NCGI forward greedy feature selection algorithms is both $O(m^2 n^2)$.

In HANGI and HANCGI, two parameters, $\sigma$ and $\beta$, are present. Parameter $\sigma$ controls the neighborhood radius, which determines the granularity of the neighborhood particles. The parameter $\beta$ is a threshold that stops the algorithm when the reduction of the GI is less than a particular value. Theoretically, the optimal values for these two parameters should be searched from the entire range of the dataset's space. Fortunately, as discussed in *Hu et al. (2008, 2011)*, for algorithms with two parameters, such as the neighborhood rough set model, it is possible to approximate the optimal performance of the algorithm if one parameter is fixed at a particular value and the optimal value of the other parameter is searched across the entire space. Since the meaning of the same-sized evaluation metric in different algorithms is not the same, in this case, all $\beta$ values in all algorithms are set to 0. This means that adding a new feature will not lead to any improvement. Based on this, in the experimental analysis section, the value of parameter $\beta$ is set as a constant 0, and the optimal value for the neighborhood radius $\sigma$ is searched within the interval [0, 1], with a step size of 0.025.

## EXPERIMENTAL ANALYSIS AND DISCUSSION

In this section, we conduct experiments to validate the effectiveness and stability of the proposed methods. We select four classic feature importance evaluation metrics based on NRS to form corresponding forward greedy feature selection algorithms: Neighborhood Rough Set Dependency (HANRS) (*Hu et al., 2008*), Neighborhood Entropy (HANRE) (*Hu et al., 2011*), Neighborhood Discrimination Index (HANDI) (*Wang et al., 2018*), and Neighborhood Self-Information (HANSI) (*Wang et al., 2019b*). We compare these algorithms with the two proposed methods. The stopping parameter $\beta = 0$ is employed as the termination condition for these algorithms.

All the datasets are sourced from the UCI Machine Learning Repository, and their specific descriptions are provided in Table 1. Where "Continuous" and "Categorical" represent the number of continuous and categorical features in each dataset. Before feature selection, all attributes are normalized to the interval [0, 1], and missing values are filled using the mean.

We compare the selected feature count and the corresponding classification accuracy to evaluate the algorithms' performance comprehensively. We employ four classical classifiers, support vector classifier (SVC), K-nearest neighbors (KNN), Extreme Gradient Boosting (XGBoost), and artificial neural network (ANN), to assess the performance of these feature selection algorithms. Since our primary focus is evaluating the feature selection algorithms, default parameter settings are used for SVC and ANN from the scikit-learn library. XGBoost also uses default parameters. For the KNN classifier, K is set to 3.

**Table 1 Description of datasets.**

| Datasets | Samples | Features | Continuous | Categorical | Classes |
|---|---|---|---|---|---|
| Anneal | 798 | 19 | 1 | 18 | 5 |
| Arrhythmia | 452 | 263 | 32 | 231 | 13 |
| Autos | 205 | 27 | 5 | 22 | 6 |
| Breast-cancer | 286 | 10 | 1 | 9 | 2 |
| DARWIN | 174 | 452 | 429 | 23 | 2 |
| Dermatology | 366 | 35 | 1 | 34 | 6 |
| HillValley | 606 | 101 | 101 | 0 | 2 |
| Horse_colic | 300 | 28 | 2 | 26 | 2 |
| Ionosphere | 351 | 34 | 33 | 1 | 2 |
| Musk1 | 476 | 169 | 85 | 84 | 2 |
| Parkinsons | 195 | 24 | 23 | 1 | 2 |
| Sonar | 208 | 61 | 61 | 0 | 2 |
| Spambase | 4,601 | 59 | 3 | 56 | 2 |
| Toxicity | 171 | 1,204 | 857 | 347 | 2 |
| Voting_records | 434 | 17 | 1 | 16 | 2 |
| Wine | 178 | 14 | 12 | 2 | 3 |

**Note:**
Continuous and categorical respectively represent the number of continuous and categorical features in each dataset.

Ten-fold cross-validation is employed to perform feature selection on these datasets. Specifically, for a given neighborhood radius $\sigma$ stopping parameter $\beta$ and a dataset, the dataset is randomly divided into ten parts, with nine parts used as the training set and one used as the test set. During the training phase, feature selection is performed on the training set to identify an optimal feature subset. The optimal feature subset is then used to extract a sub-dataset from the original dataset. During the testing phase, ten-fold cross-validation is applied to the sub-dataset, computing the accuracy of the four classifiers. Finally, the mean of the output accuracy values obtained from four classifiers serves as the ultimate evaluation metric, providing a comprehensive assessment of feature selection effectiveness across the entire dataset.

## Training parameters

In NRS-based models, the size of the neighborhood granule significantly impacts the model results. Determining the neighborhood granule's size is essential to achieve optimal experimental outcomes. Thus, we employ ten-fold cross-validation with a step size of 0.025 in the range (0, 1) (*Wang et al., 2019b*) to obtain the optimal neighborhood radius parameter $\sigma$ for each algorithm. The search range in the "Spambase" dataset is (0, 0.225). Subsequently, we use four datasets and one algorithm to illustrate the selection process. Figure 8 displays the variation of classification accuracy with changing neighborhood radius for different datasets, using NGI as the evaluation metric.

Evidently, the neighborhood radius has a pronounced impact on classification accuracy. As the parameter changes, the four datasets exhibit varying accuracy levels in all classifiers. We select the radius that corresponds to relatively higher accuracy in all classifiers as the

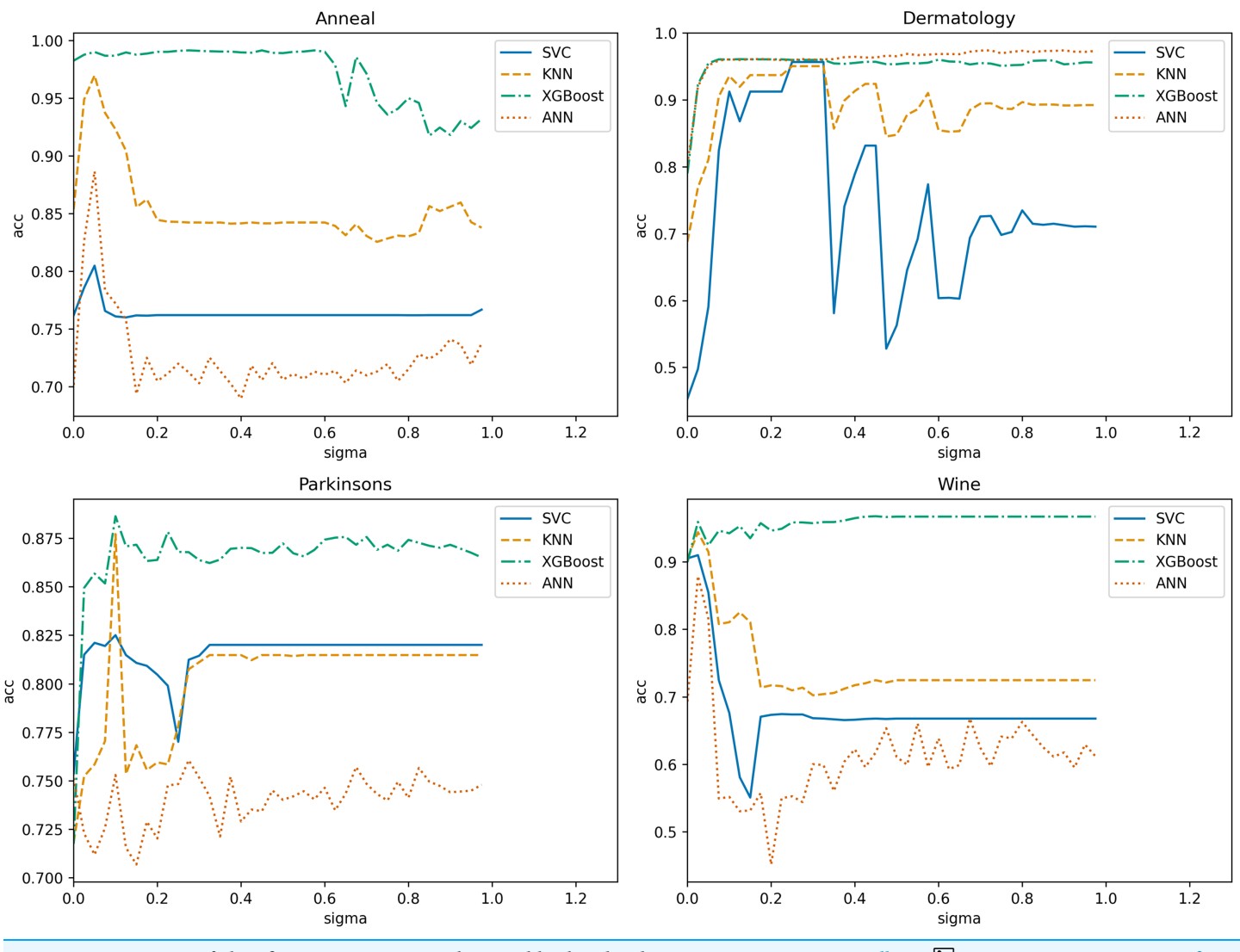

**Figure 8 Variation of classification accuracies with a neighborhood radius.**

optimal radius. For instance, in the "Anneal" dataset, $\sigma = 0.05$ is deemed the optimal neighborhood radius. Using the same training methodology, we determine the optimal neighborhood radius for each algorithm on various datasets, as presented in Table 2. In subsequent comparisons of algorithm performance, the neighborhood radius parameters are set based on this table.

In Table 2, the first column represents the dataset name, and each subsequent column header corresponds to the algorithm's name. The values inside the table indicate the optimal neighborhood radius for each algorithm.

It is important to note that for the "Voting_records" dataset, HANRS cannot select features at any neighborhood radius. Therefore, we set it to the minimum value of 0.025 for subsequent comparisons.

**Table 2 Optimal neighborhood radius parameters.**

| Datasets | HANRS | HANMI | HANDI | HANSI | HANGI | HANCGI |
|---|---|---|---|---|---|---|
| Anneal | 0.05 | 0.725 | 0.05 | 0.05 | 0.05 | 0.05 |
| Arrhythmia | 0.35 | 0.95 | 0.275 | 0.875 | 0.275 | 0.3 |
| Autos | 0.1 | 0.6 | 0.075 | 0.15 | 0.125 | 0.125 |
| Breast-cancer | 0.4 | 0.025 | 0.725 | 0.4 | 0.125 | 0.25 |
| DARWIN | 0.075 | 0.925 | 0.975 | 0.85 | 0.975 | 0.025 |
| Dermatology | 0.175 | 0.025 | 0.425 | 0.225 | 0.275 | 0.575 |
| HillValley | 0.225 | 0.525 | 0.2 | 0.425 | 0.425 | 0.425 |
| Horse_colic | 0.325 | 0.825 | 0.15 | 0.325 | 0.225 | 0.575 |
| Ionosphere | 0.175 | 0.525 | 0.175 | 0.2 | 0.15 | 0.125 |
| Musk1 | 0.65 | 0.95 | 0.975 | 0.65 | 0.95 | 0.425 |
| Parkinsons | 0.1 | 0.375 | 0.1 | 0.1 | 0.1 | 0.375 |
| Sonar | 0.475 | 0.775 | 0.55 | 0.525 | 0.45 | 0.325 |
| Spambase | 0.175 | 0.15 | 0.1 | 0.175 | 0.15 | 0.125 |
| Toxicity | 0.975 | 0.075 | 0.05 | 0.95 | 0.075 | 0.925 |
| Voting_records | 0.025 | 0.025 | 0.875 | 0.25 | 0.875 | 0.425 |
| Wine | 0.025 | 0.95 | 0.025 | 0.025 | 0.025 | 0.05 |

**Note:**
The underlines represent that the results corresponding to all neighborhood radii under this algorithm are exactly the same.

## Evaluation of feature validity

In the context of classification problems, feature selection algorithms aim to extract the most representative and discriminative features from the original feature set, creating a more compact subset. Constructing a classification model using the selected feature subset, achieving higher accuracy indicates that these features are more effective for the classification task on the given data. Based on the optimal neighborhood radius, the feature selection algorithms (HANRS, HANRE, HANDI, HANSI, HANGI, and HANCGI) were applied to 16 datasets, and the number of features selected is presented in Table 3. Where "Original" denotes the original dataset's number of features, each subsequent column represents the average number of features selected by each algorithm over ten runs. Underscored numbers indicate the fewest selected features relative to other algorithms. Notably, HANRS did not select features in the "Voting_records" dataset and, therefore, is not included in the comparison.

Comparing the number of selected features in Table 3, we observe that HANGI and HANCGI successfully achieve feature reduction. There is no significant difference in the average number of features reduced among the six algorithms. HANMI shows the strongest reduction capability, while HANSI demonstrates the weakest. Across the 16 datasets, the average number of features was reduced by HANGI to 18, ranking fifth on average among the six algorithms. HANCGI reduces the average number of features to 11, ranking second on average among the six algorithms.

Next, we employ SVC, KNN, XGBoost, and ANN to train the selected feature subsets and compare their classification accuracies, as presented in Tables 4–7. Table 8 presents

**Table 3 Number of selected features.**

| Datasets | Original | HANRS | HANMI | HANDI | HANSI | HANGI | HANCGI |
|---|---|---|---|---|---|---|---|
| Anneal | 19 | 7.20 | 3.20 | 8.50 | 8.00 | 7.70 | 7.60 |
| Arrhythmia | 263 | 34.00 | 6.80 | 17.10 | 115.60 | 16.40 | 28.20 |
| Autos | 27 | 9.10 | 1.10 | 7.20 | 10.00 | 9.10 | 8.60 |
| Breast-cancer | 10 | 1.00 | 1.00 | 5.80 | 1.00 | 8.10 | 1.00 |
| DARWIN | 452 | 4.80 | 9.60 | 36.90 | 48.60 | 45.40 | 3.40 |
| Dermatology | 35 | 10.10 | 1.00 | 12.60 | 11.20 | 10.30 | 18.80 |
| HillValley | 101 | 5.00 | 1.90 | 2.40 | 2.80 | 18.60 | 2.40 |
| Horse_colic | 28 | 16.00 | 3.00 | 10.90 | 15.90 | 13.30 | 1.00 |
| Ionosphere | 34 | 10.80 | 2.00 | 8.90 | 11.90 | 8.10 | 7.70 |
| Musk1 | 169 | 34.90 | 9.50 | 64.60 | 35.50 | 63.70 | 16.60 |
| Parkinsons | 24 | 4.00 | 1.30 | 4.00 | 4.40 | 4.00 | 4.20 |
| Sonar | 61 | 21.70 | 3.00 | 24.90 | 25.60 | 18.40 | 11.10 |
| Spambase | 59 | 50.10 | 58.00 | 36.20 | 50.10 | 49.00 | 44.40 |
| Toxicity | 1,204 | 6.60 | 4.90 | 4.00 | 6.60 | 4.90 | 1.30 |
| Voting_records | 17 | 0.00 | 1.00 | 10.60 | 10.80 | 13.00 | 9.80 |
| Wine | 14 | 2.90 | 2.00 | 2.90 | 2.90 | 2.90 | 3.00 |
| Mean | 157.3125 | 13.64 | 6.83 | 16.09 | 22.56 | 18.31 | 10.57 |

**Note:**
Underscored numbers indicate the fewest selected features relative to other algorithms.

the mean accuracy of each dataset across the four classifiers. In these tables, underscored numbers indicate the best classification accuracy achieved through feature reduction relative to other algorithms.

Tables 4–8 demonstrate that the HANGI and HANCGI algorithms proposed in this article effectively improve classification accuracy. On average, HANGI improved classification accuracy on 14 datasets across the four classifiers, with improvements exceeding 10% on four datasets and an average accuracy improvement of 7%. HANCGI improved classification accuracy on 12 datasets, with improvements exceeding 10% on five datasets and an average accuracy improvement of 6.6%. Among the classifiers, XGBoost showed the smallest improvement in classification accuracy, with an average accuracy decrease of 0.8% for the selected features by HANCGI. This is because XGBoost not only acts as a classifier but also is an embedded feature selection model, automatically selecting features during the classification process to enhance accuracy. The results in Table 6 indicate that XGBoost's feature selection results are similar to those of HANGI and HANCGI, with no significant difference in overall classification accuracy. In some datasets, the proposed algorithms have improved the classification accuracy of XGBoost by removing redundant and irrelevant features. For example, on the Toxicity dataset, both HANGI and HANCGI improved classification accuracy on XGBoost.

In the case of the 16 datasets, most of them showed improved classification accuracy after feature selection using the six feature selection algorithms. Although HANMI exhibited the strongest feature reduction capability, it had the poorest performance in terms of classification accuracy across the four classifiers. This suggests that HANMI might

**Table 4 Average accuracy on SVC.**

| Datasets | Original | HANRS | HANMI | HANDI | HANSI | HANGI | HANCGI |
|---|---|---|---|---|---|---|---|
| Anneal | 0.7619 | 0.7998 | 0.8344 | 0.8047 | 0.8046 | 0.8047 | 0.7997 |
| Arrhythmia | 0.6106 | 0.5961 | 0.5839 | 0.6306 | 0.6024 | 0.6534 | 0.6454 |
| Autos | 0.4105 | 0.3385 | 0.4418 | 0.4428 | 0.3665 | 0.4009 | 0.3997 |
| Breast-cancer | 0.6873 | 0.7622 | 0.7622 | 0.7449 | 0.7622 | 0.7405 | 0.7622 |
| DARWIN | 0.4647 | 0.7611 | 0.7178 | 0.5897 | 0.5654 | 0.5654 | 0.7631 |
| Dermatology | 0.7297 | 0.9520 | 0.3240 | 0.9209 | 0.9550 | 0.9566 | 0.9758 |
| HillValley | 0.5100 | 0.4774 | 0.4751 | 0.4777 | 0.4764 | 0.4745 | 0.4759 |
| Horse_colic | 0.6567 | 0.6730 | 0.6823 | 0.6613 | 0.6740 | 0.6637 | 0.7667 |
| Ionosphere | 0.9344 | 0.9378 | 0.8766 | 0.9387 | 0.9432 | 0.9312 | 0.9359 |
| Musk1 | 0.7737 | 0.8075 | 0.7557 | 0.8482 | 0.8100 | 0.8382 | 0.7405 |
| Parkinsons | 0.8100 | 0.8250 | 0.7970 | 0.8250 | 0.8225 | 0.8250 | 0.8164 |
| Sonar | 0.6395 | 0.8104 | 0.6921 | 0.8095 | 0.7961 | 0.8108 | 0.7645 |
| Spambase | 0.9538 | 0.9940 | 0.9946 | 0.9940 | 0.9940 | 0.9940 | 0.9939 |
| Toxicity | 0.6500 | 0.6717 | 0.6717 | 0.6729 | 0.6717 | 0.6729 | 0.6711 |
| Voting_records | 0.9585 | 0.0000 | 0.6154 | 0.9636 | 0.9621 | 0.9641 | 0.9344 |
| Wine | 0.6810 | 0.8610 | 0.5833 | 0.9096 | 0.9096 | 0.9096 | 0.8576 |
| Mean | 0.7020 | 0.7042 | 0.6755 | 0.7646 | 0.7572 | 0.7628 | 0.7689 |

Note:
Underscored numbers indicate the best classification accuracy achieved through feature reduction relative to other algorithms.

**Table 5 Average accuracy on KNN.**

| Datasets | Original | HANRS | HANMI | HANDI | HANSI | HANGI | HANCGI |
|---|---|---|---|---|---|---|---|
| Anneal | 0.8622 | 0.9698 | 0.9123 | 0.9750 | 0.9800 | 0.9696 | 0.9639 |
| Arrhythmia | 0.6106 | 0.6012 | 0.5583 | 0.6226 | 0.6000 | 0.6227 | 0.6311 |
| Autos | 0.3964 | 0.6976 | 0.6117 | 0.6955 | 0.6800 | 0.7050 | 0.6933 |
| Breast-cancer | 0.7326 | 0.7622 | 0.7622 | 0.7533 | 0.7622 | 0.7580 | 0.7622 |
| DARWIN | 0.6690 | 0.7978 | 0.7867 | 0.7561 | 0.7168 | 0.7322 | 0.7364 |
| Dermatology | 0.9040 | 0.9418 | 0.2965 | 0.9437 | 0.9531 | 0.9505 | 0.9734 |
| HillValley | 0.5663 | 0.5515 | 0.5077 | 0.5097 | 0.5469 | 0.5400 | 0.5306 |
| Horse_colic | 0.6067 | 0.6473 | 0.7343 | 0.6480 | 0.6427 | 0.6727 | 0.8233 |
| Ionosphere | 0.8348 | 0.8905 | 0.8573 | 0.8892 | 0.8715 | 0.8908 | 0.9034 |
| Musk1 | 0.7905 | 0.7988 | 0.7809 | 0.8354 | 0.8117 | 0.8288 | 0.7799 |
| Parkinsons | 0.7984 | 0.8771 | 0.7532 | 0.8771 | 0.8504 | 0.8771 | 0.8076 |
| Sonar | 0.5952 | 0.8384 | 0.6794 | 0.8357 | 0.8280 | 0.8372 | 0.7834 |
| Spambase | 0.9538 | 0.9989 | 0.9985 | 0.9988 | 0.9989 | 0.9988 | 0.9989 |
| Toxicity | 0.5275 | 0.6523 | 0.5831 | 0.6038 | 0.5398 | 0.6201 | 0.6688 |
| Voting_records | 0.9469 | 0.0000 | 0.6086 | 0.9426 | 0.9339 | 0.9393 | 0.9008 |
| Wine | 0.7209 | 0.8962 | 0.5992 | 0.9436 | 0.9436 | 0.9436 | 0.9124 |
| Mean | 0.7197 | 0.7451 | 0.6894 | 0.8019 | 0.7912 | 0.8054 | 0.8043 |

Note:
Underscored numbers indicate the best classification accuracy achieved through feature reduction relative to other algorithms.

**Table 6 Average accuracy on XGBoost.**

| Datasets | Original | HANRS | HANMI | HANDI | HANSI | HANGI | HANCGI |
|---|---|---|---|---|---|---|---|
| Anneal | 0.9900 | 0.9913 | 0.9461 | 0.9903 | 0.9900 | 0.9898 | 0.9815 |
| Arrhythmia | 0.7342 | 0.6891 | 0.5483 | 0.7056 | 0.7299 | 0.6971 | 0.7295 |
| Autos | 0.6660 | 0.6934 | 0.4899 | 0.6750 | 0.6954 | 0.6944 | 0.6820 |
| Breast-cancer | 0.6701 | 0.7623 | 0.7623 | 0.7014 | 0.7623 | 0.6772 | 0.7623 |
| DARWIN | 0.8428 | 0.8057 | 0.7900 | 0.8525 | 0.8289 | 0.8317 | 0.7681 |
| Dermatology | 0.9565 | 0.9454 | 0.3307 | 0.9576 | 0.9556 | 0.9598 | 0.9590 |
| HillValley | 0.6123 | 0.5925 | 0.5273 | 0.5539 | 0.6275 | 0.6044 | 0.6062 |
| Horse_colic | 0.8467 | 0.8357 | 0.7737 | 0.8240 | 0.8417 | 0.8457 | 0.8433 |
| Ionosphere | 0.9089 | 0.9192 | 0.8339 | 0.9307 | 0.9192 | 0.9288 | 0.9189 |
| Musk1 | 0.7778 | 0.7880 | 0.7418 | 0.7690 | 0.7779 | 0.7775 | 0.7556 |
| Parkinsons | 0.8611 | 0.8863 | 0.7178 | 0.8863 | 0.8791 | 0.8863 | 0.7817 |
| Sonar | 0.7360 | 0.7433 | 0.6513 | 0.7414 | 0.7531 | 0.7723 | 0.6887 |
| Spambase | 0.9501 | 0.9501 | 0.9501 | 0.9501 | 0.9501 | 0.9501 | 0.9501 |
| Toxicity | 0.5618 | 0.6614 | 0.6094 | 0.6434 | 0.6567 | 0.6191 | 0.6637 |
| Voting_records | 0.9585 | 0.0000 | 0.6152 | 0.9626 | 0.9569 | 0.9615 | 0.9299 |
| Wine | 0.9667 | 0.9460 | 0.6397 | 0.9589 | 0.9589 | 0.9589 | 0.9192 |
| Mean | 0.8149 | 0.7631 | 0.6830 | 0.8189 | 0.8302 | 0.8222 | 0.8087 |

**Note:**
Underscored numbers indicate the best classification accuracy achieved through feature reduction relative to other algorithms.

**Table 7 Average accuracy on ANN.**

| Datasets | Original | HANRS | HANMI | HANDI | HANSI | HANGI | HANCGI |
|---|---|---|---|---|---|---|---|
| Anneal | 0.7379 | 0.8557 | 0.8205 | 0.9041 | 0.9171 | 0.8865 | 0.8989 |
| Arrhythmia | 0.6106 | 0.5711 | 0.5471 | 0.5904 | 0.5737 | 0.5839 | 0.6156 |
| Autos | 0.2286 | 0.4590 | 0.3219 | 0.4760 | 0.4905 | 0.4135 | 0.4475 |
| Breast-cancer | 0.6942 | 0.7623 | 0.7623 | 0.6964 | 0.7623 | 0.7106 | 0.7623 |
| DARWIN | 0.8203 | 0.6229 | 0.5761 | 0.6946 | 0.7357 | 0.7210 | 0.5719 |
| Dermatology | 0.9699 | 0.9518 | 0.3348 | 0.9587 | 0.9565 | 0.9609 | 0.9703 |
| HillValley | 0.5775 | 0.6203 | 0.5155 | 0.5615 | 0.6456 | 0.6205 | 0.6427 |
| Horse_colic | 0.5767 | 0.6997 | 0.6753 | 0.6717 | 0.7013 | 0.6530 | 0.7587 |
| Ionosphere | 0.9260 | 0.9269 | 0.7953 | 0.9233 | 0.9300 | 0.9165 | 0.9215 |
| Musk1 | 0.7738 | 0.7245 | 0.7101 | 0.7324 | 0.7243 | 0.7400 | 0.6736 |
| Parkinsons | 0.7129 | 0.7466 | 0.7487 | 0.7422 | 0.7424 | 0.7529 | 0.7558 |
| Sonar | 0.6550 | 0.6985 | 0.6965 | 0.6959 | 0.6903 | 0.7144 | 0.6728 |
| Spambase | 0.9776 | 0.9688 | 0.9708 | 0.9670 | 0.9680 | 0.9676 | 0.9664 |
| Toxicity | 0.4794 | 0.6656 | 0.6552 | 0.6412 | 0.6649 | 0.6399 | 0.6708 |
| Voting_records | 0.9608 | 0.0000 | 0.6152 | 0.9635 | 0.9608 | 0.9633 | 0.9343 |
| Wine | 0.5471 | 0.7879 | 0.4799 | 0.8788 | 0.8823 | 0.8788 | 0.8113 |
| Mean | 0.7030 | 0.6913 | 0.6391 | 0.7561 | 0.7716 | 0.7577 | 0.7546 |

**Note:**
Underscored numbers indicate the best classification accuracy achieved through feature reduction relative to other algorithms.

**Table 8 Average accuracy on four classifiers.**

| Datasets | Original | HANRS | HANMI | HANDI | HANSI | HANGI | HANCGI |
|---|---|---|---|---|---|---|---|
| Anneal | 0.8380 | 0.9042 | 0.8783 | 0.9185 | 0.9229 | 0.9127 | 0.9110 |
| Arrhythmia | 0.6415 | 0.6144 | 0.5594 | 0.6373 | 0.6265 | 0.6393 | 0.6554 |
| Autos | 0.4254 | 0.5471 | 0.4664 | 0.5723 | 0.5581 | 0.5535 | 0.5556 |
| Breast-cancer | 0.6961 | 0.7623 | 0.7623 | 0.7240 | 0.7623 | 0.7216 | 0.7623 |
| DARWIN | 0.6992 | 0.7469 | 0.7177 | 0.7232 | 0.7117 | 0.7126 | 0.7099 |
| Dermatology | 0.8900 | 0.9478 | 0.3215 | 0.9452 | 0.9551 | 0.9570 | 0.9696 |
| HillValley | 0.5665 | 0.5604 | 0.5064 | 0.5257 | 0.5741 | 0.5599 | 0.5638 |
| Horse_colic | 0.6717 | 0.7139 | 0.7164 | 0.7013 | 0.7149 | 0.7088 | 0.7980 |
| Ionosphere | 0.9010 | 0.9186 | 0.8408 | 0.9205 | 0.9160 | 0.9168 | 0.9199 |
| Musk1 | 0.7790 | 0.7797 | 0.7472 | 0.7962 | 0.7810 | 0.7961 | 0.7374 |
| Parkinsons | 0.7956 | 0.8338 | 0.7542 | 0.8326 | 0.8236 | 0.8353 | 0.7904 |
| Sonar | 0.6564 | 0.7726 | 0.6798 | 0.7706 | 0.7669 | 0.7836 | 0.7273 |
| Spambase | 0.9588 | 0.9779 | 0.9785 | 0.9775 | 0.9777 | 0.9776 | 0.9773 |
| Toxicity | 0.5547 | 0.6628 | 0.6298 | 0.6403 | 0.6333 | 0.6380 | 0.6686 |
| Voting_records | 0.9562 | 0.0000 | 0.6136 | 0.9581 | 0.9534 | 0.9570 | 0.9248 |
| Wine | 0.7289 | 0.8728 | 0.5755 | 0.9227 | 0.9236 | 0.9227 | 0.8751 |
| Mean | 0.7349 | 0.7259 | 0.6717 | 0.7854 | 0.7876 | 0.7870 | 0.7842 |

**Note:**
Underscored numbers indicate the best classification accuracy achieved through feature reduction relative to other algorithms.

have discarded some crucial features during the feature selection process, resulting in lower classification accuracy.

HANDI, HANSI, the proposed HANGI, and HANCGI all showed relatively similar classification accuracy for the selected features across the four classifiers, but there were significant differences in some datasets. For example, on the DARWIN dataset, the features selected by HANCGI performed significantly better with SVC than HANDI, and the number of features selected by HANCGI was also much fewer than HANDI. On the "Parkinsons" dataset, despite a similar number of selected features between HANCGI and HANSI, HANCGI exhibited significantly lower classification accuracy with XGBoost.

It is worth mentioning that on the "Spambase" dataset, the selected features by all six feature selection algorithms achieved the same accuracy with XGBoost as the original dataset, even though the number of selected features differed among the algorithms. This suggests that all six feature selection algorithms got what XGBoost considered the crucial features, with HANDI selecting the fewest features and HANGI and HANCGI following closely behind.

Through comprehensive analysis of the experimental results, it is evident that the proposed methods often select fewer features while maintaining or improving classification accuracy. This suggests that the proposed methods can effectively eliminate more redundant attributes. Next, we evaluate the statistical significance of the performance differences among the six algorithms for feature selection through hypothesis testing. We

**Table 9 Rank of the six algorithms with the average accuracy on four classifiers.**

| Datasets | HANRS | HANMI | HANDI | HANSI | HANGI | HANCGI |
|---|---|---|---|---|---|---|
| Anneal | 2.00 | 1.00 | 5.00 | 6.00 | 4.00 | 3.00 |
| Arrhythmia | 2.00 | 1.00 | 4.00 | 3.00 | 5.00 | 6.00 |
| Autos | 2.00 | 1.00 | 6.00 | 5.00 | 3.00 | 4.00 |
| Breast-cancer | 4.50 | 4.50 | 2.00 | 4.50 | 1.00 | 4.50 |
| DARWIN | 6.00 | 4.00 | 5.00 | 2.00 | 3.00 | 1.00 |
| Dermatology | 3.00 | 1.00 | 2.00 | 4.00 | 5.00 | 6.00 |
| HillValley | 4.00 | 1.00 | 2.00 | 6.00 | 3.00 | 5.00 |
| Horse_colic | 3.00 | 5.00 | 1.00 | 4.00 | 2.00 | 6.00 |
| Ionosphere | 4.00 | 1.00 | 6.00 | 2.00 | 3.00 | 5.00 |
| Musk1 | 3.00 | 2.00 | 6.00 | 4.00 | 5.00 | 1.00 |
| Parkinsons | 5.00 | 1.00 | 4.00 | 3.00 | 6.00 | 2.00 |
| Sonar | 5.00 | 1.00 | 4.00 | 3.00 | 6.00 | 2.00 |
| Spambase | 5.00 | 6.00 | 2.00 | 4.00 | 3.00 | 1.00 |
| Toxicity | 5.00 | 1.00 | 4.00 | 2.00 | 3.00 | 6.00 |
| Voting_records | 1.00 | 2.00 | 6.00 | 4.00 | 5.00 | 3.00 |
| Wine | 2.00 | 1.00 | 4.50 | 6.00 | 4.50 | 3.00 |
| Mean | 3.44 | 2.03 | 3.97 | 4.03 | 3.91 | 3.62 |

first examine whether there are significant differences among the six algorithms on these datasets, utilizing the Friedman test statistic (*Friedman, 1940*):

$$\chi^2 = \frac{12n}{k(k+1)} \left( \sum_{i=1}^{k} r_i^2 - \frac{k(k+1)^2}{4} \right) \tag{17}$$

$$F = \frac{(n-1)\chi^2}{n(k-1) - \chi^2} \tag{18}$$

where $r_i$ represents the average rank of the algorithm, $n$ denotes the number of datasets, and $k$ represents the number of algorithms. The random variable "F" follows an F-distribution with degrees of freedom $k-1$ and $(k-1)(n-1)$. The critical value of the F-distribution at a significance level $\alpha$ can be obtained by invoking the subroutine 'scipy.stats. f.ppf(1-α, n-1, (k-1)*(n-1))' in Python 3.9. Thus, when $\alpha = 0.05$, we obtain the critical value F(5,75) = 2.337. If the performance of the six algorithms is similar, the value of the Friedman statistic should not exceed the critical value F(5,75). Otherwise, there would be a significant difference in the feature selection performance among these six algorithms.

Table 9 displays the performance ranking order of the six algorithms' selected features across the four classifiers arranged in ascending order. More significant numbers indicate better classification performance. According to the Friedman test statistic, we can obtain that F = 2.507 > 2.337 for the four classifiers. Evidently, there is a significant difference among the six algorithms on the four classifiers.

At this point, further *post hoc* tests are necessary to examine the differences among the six algorithms. The *post hoc* test employed here is the Nemenyi test. This statistical test

requires determining the critical distance between average ranking values, defined by the following formula:

$$CD_\alpha = q_\alpha \sqrt{\frac{k(k+1)}{6N}} \qquad (19)$$

where $q_\alpha$ is the critical tabulated value for this test. From *Demšar (2006)*, we can obtain that $q_{0.05} = 2.850$ when the number of algorithms is 6 and $\alpha = 0.05$. It follows from the above formula that $CD_{0.05} = 1.885(k = 6, n = 16)$. If the corresponding average rank distance is greater than the critical distance $CD_{0.05}$, it indicates a significant difference between the two algorithms.

It is easy to observe from Table 9 that the average ranking distance between HANSI and HANDI compared to HANMI is bigger than 1.885, indicating a significant difference in performance. However, the proposed HANGI and HANCGI, compared to the other four algorithms, have an average ranking distance of less than 1.885. This suggests that the algorithms proposed in the article do not exhibit a significant difference in average performance compared to the other four algorithms across the four classifiers.

## Evaluation of algorithm stability

The stability of a feature selection algorithm refers to its ability to produce consistent or similar feature selection results when the dataset undergoes certain perturbations, such as removing or adding some samples. To discuss algorithm stability, we simulate removing a portion of samples. The procedure is as follows: First, the samples are randomly divided into ten subsets. In each iteration, nine subsets are chosen, and the feature selection algorithm is applied to obtain an optimal feature subset. This process is repeated ten times, resulting in 10 feature subsets. Features that appear at least five times out of the ten subsets are selected using a majority voting principle to form the final feature subset. Table 10 shows the number of features appearing at least once in each dataset's ten feature selection results. Table 11 shows the number of features that appear at least five times in the ten feature selection results. Table 12 presents the ratio of feature numbers before and after voting, where a higher ratio indicates higher stability of the corresponding feature selection algorithm.

If a feature repeats occurrences across the ten feature subsets, it indicates a high level of reproducibility for that feature. The features that appear frequently in the ten subsets are selected through voting. If the final selected feature subset contains more features, it suggests a higher degree of algorithm stability. Table 12 presents the stability performance of the six algorithms across different datasets, with the most stable algorithm for each dataset marked with an underline.

Table 12 shows that HANSI has the highest stability, reaching 0.64, followed by HANCGI, with a stability of 0.59. The remaining algorithms have relatively similar stability, with HANMI showing the poorest stability. This difference in stability can be attributed to the fact that HANSI and HANCGI assess feature importance based on the class distribution within the class neighborhood range. However, other algorithms evaluate

**Table 10 Number of features before voting.**

| Datasets | HANRS | HANMI | HANDI | HANSI | HANGI | HANCGI |
|---|---|---|---|---|---|---|
| Anneal | 11 | 4 | 11 | 9 | 13 | 10 |
| Arrhythmia | 107 | 30 | 59 | 186 | 52 | 110 |
| Autos | 18 | 3 | 17 | 16 | 16 | 17 |
| Breast-cancer | 1 | 1 | 8 | 1 | 9 | 1 |
| DARWIN | 16 | 39 | 155 | 199 | 182 | 18 |
| Dermatology | 18 | 2 | 18 | 25 | 18 | 23 |
| HillValley | 12 | 9 | 19 | 4 | 67 | 3 |
| Horse_colic | 23 | 6 | 20 | 22 | 24 | 1 |
| Ionosphere | 20 | 4 | 19 | 28 | 21 | 19 |
| Musk1 | 98 | 34 | 140 | 104 | 142 | 61 |
| Parkinsons | 4 | 2 | 4 | 7 | 4 | 5 |
| Sonar | 49 | 8 | 51 | 55 | 49 | 40 |
| Spambase | 56 | 58 | 47 | 56 | 56 | 52 |
| Toxicity | 15 | 25 | 18 | 15 | 21 | 5 |
| Voting_records | 0 | 1 | 16 | 13 | 15 | 15 |
| Wine | 5 | 5 | 3 | 3 | 3 | 8 |

**Table 11 Number of features after voting.**

| Datasets | HANRS | HANMI | HANDI | HANSI | HANGI | HANCGI |
|---|---|---|---|---|---|---|
| Anneal | 7 | 3 | 9 | 8 | 8 | 8 |
| Arrhythmia | 27 | 4 | 12 | 122 | 12 | 19 |
| Autos | 11 | 1 | 6 | 9 | 10 | 9 |
| Breast-cancer | 1 | 1 | 5 | 1 | 8 | 1 |
| DARWIN | 4 | 6 | 19 | 29 | 24 | 2 |
| Dermatology | 10 | 2 | 13 | 13 | 9 | 19 |
| HillValley | 5 | 1 | 1 | 3 | 12 | 3 |
| Horse_colic | 19 | 3 | 13 | 18 | 13 | 1 |
| Ionosphere | 11 | 2 | 8 | 11 | 8 | 7 |
| Musk1 | 30 | 7 | 59 | 29 | 55 | 12 |
| Parkinsons | 4 | 1 | 4 | 4 | 4 | 5 |
| Sonar | 25 | 3 | 26 | 24 | 16 | 6 |
| Spambase | 51 | 58 | 38 | 51 | 51 | 47 |
| Toxicity | 6 | 3 | 1 | 6 | 3 | 1 |
| Voting_records | 0 | 1 | 10 | 12 | 14 | 12 |
| Wine | 3 | 1 | 3 | 3 | 3 | 3 |

feature importance based on the class distribution within the sample neighborhood. When some samples are perturbed or removed, it inevitably affects the class distribution within their respective neighborhood range, thus influencing the assessment of feature importance.

**Table 12  Ratio of feature numbers before and after voting.**

| Datasets | HANRS | HANMI | HANDI | HANSI | HANGI | HANCGI |
|---|---|---|---|---|---|---|
| Anneal | 0.64 | 0.75 | 0.82 | 0.89 | 0.62 | 0.80 |
| Arrhythmia | 0.25 | 0.13 | 0.20 | 0.66 | 0.23 | 0.17 |
| Autos | 0.61 | 0.33 | 0.35 | 0.56 | 0.63 | 0.53 |
| Breast-cancer | 1.00 | 1.00 | 0.63 | 1.00 | 0.89 | 1.00 |
| DARWIN | 0.25 | 0.15 | 0.12 | 0.15 | 0.13 | 0.11 |
| Dermatology | 0.56 | 1.00 | 0.72 | 0.52 | 0.50 | 0.83 |
| HillValley | 0.42 | 0.11 | 0.05 | 0.75 | 0.18 | 1.00 |
| Horse_colic | 0.83 | 0.50 | 0.65 | 0.82 | 0.54 | 1.00 |
| Ionosphere | 0.55 | 0.50 | 0.42 | 0.39 | 0.38 | 0.37 |
| Musk1 | 0.31 | 0.21 | 0.42 | 0.28 | 0.39 | 0.20 |
| Parkinsons | 1.00 | 0.50 | 1.00 | 0.57 | 1.00 | 1.00 |
| Sonar | 0.51 | 0.38 | 0.51 | 0.44 | 0.33 | 0.15 |
| Spambase | 0.91 | 1.00 | 0.81 | 0.91 | 0.91 | 0.90 |
| Toxicity | 0.40 | 0.12 | 0.06 | 0.40 | 0.14 | 0.20 |
| Voting_records | 0.00 | 1.00 | 0.63 | 0.92 | 0.93 | 0.80 |
| Wine | 0.60 | 0.20 | 1.00 | 1.00 | 1.00 | 0.38 |
| Mean | 0.55 | 0.49 | 0.52 | 0.64 | 0.55 | 0.59 |

**Note:**
Underlined numbers indicate higher stability relative to other algorithms.

Algorithms that assess feature importance by considering the distribution of classes within the sample neighborhood primarily focus on the local class distribution. When there is local perturbation, it directly impacts the evaluation of feature importance, resulting in lower stability. In contrast, HANCGI and HANSI pay more attention to the class neighborhood's class distribution. When local interference occurs, it first affects the neighborhood of their respective classes, and in this process, interference is averaged out by unaffected samples within that class. Subsequently, feature importance assessment is influenced by the class neighborhood, and during this process, it is further averaged out by other unaffected classes. Therefore, these algorithms exhibit higher stability. HANCGI considers the distribution of all classes within the class neighborhood range, while HANSI only considers whether the classes within the class neighborhood are the same as the primary class. Therefore, HANSI exhibits stronger robustness to disturbances.

The experimental results above demonstrate that the algorithms proposed in this article exhibit high stability and strong feature reduction capabilities, particularly in removing redundant and irrelevant features, resulting in improved classification accuracy on most datasets. HANSI demonstrates the highest stability and achieves the best classification performance across the four classifiers, but it has the weakest feature reduction capability. On the other hand, HANGI, proposed in this article, has stronger feature reduction capabilities than HANSI, with slightly lower stability, and its selected features exhibit classification performance just 0.06% worse than HANSI on average. HANCGI boasts significantly stronger feature reduction capabilities than HANSI, with slightly less stability,

and its selected features have an average classification performance of only 0.3% worse than HANSI.

In conclusion, compared to four classical feature selection algorithms based on neighborhood rough sets, the algorithms proposed in this article outperform three and have advantages and disadvantages compared to HANSI.

## CONCLUSIONS

The assessment of feature subset importance is crucial in classification learning and feature selection. There is currently a plethora of metrics available for evaluating feature importance. The Gini index has already been proven effective in classification learning and feature selection. In this article, we introduce the Gini index into the realm of neighborhood rough sets and propose two evaluation metrics for measuring the importance of feature subsets. These two metrics combine the Gini index at the level of sample neighborhoods and class neighborhoods, respectively, to gauge the importance of feature subsets. They assess the importance of feature subsets based on the purity of class distributions within the scope of sample neighborhoods and class neighborhoods. Subsequently, we delve into the properties of these two evaluation metrics and their relationships with attributes. Leveraging the assessment of candidate features' importance relative to existing feature subsets, we put forth two greedy heuristic algorithms to eliminate redundant and irrelevant features.

To comprehensively assess the performance of the algorithms proposed in this article, we conducted comparative experiments on 16 UCI datasets spanning various domains, including industrial, food, medical, and pharmaceutical fields, with four classical feature selection algorithms based on neighborhood rough sets. The experimental results demonstrate that HANGI and HANCGI effectively remove a substantial portion of redundant and irrelevant features, leading to enhanced classification accuracy while exhibiting high stability. Across the 16 UCI datasets, the average classification accuracy improved by more than 6%, with five datasets showing an average accuracy improvement exceeding 10%.

Compared to the four classical feature selection algorithms based on neighborhood rough sets, the two proposed algorithms showed no statistically significant difference in the average classification accuracy of the selected features across the four classifiers. However, HANCGI selected fewer features while maintaining the same level of classification accuracy, indicating its superior capability to eliminate redundant and irrelevant features compared to the other four algorithms. Additionally, the algorithms proposed in this article demonstrated high stability, with performance slightly below that of HANSI.

In conclusion, the algorithms proposed in this article outperformed three classical feature selection algorithms based on neighborhood rough sets. They had their strengths and weaknesses in comparison to HANSI.

It is worth noting that in the section where we discussed properties, we explored the relationships between NGI, NCGI, and feature subsets. Our examination reveals that while the expansion of the feature subset generally corresponds to a decline in overall GI, this trend is not entirely monotonic. Hence, the optimal feature subset found by the proposed

forward greedy algorithm is a local optimum rather than a global one. Subsequent research endeavors may explore more sophisticated feature selection mechanisms, such as intelligent optimization algorithms, to ascertain globally optimal feature subsets for attaining superior results.

## ACKNOWLEDGEMENTS

During the process of completing this research, I have received encouragement, support, and assistance from many individuals, and I would like to express my heartfelt gratitude to them. First and foremost, I sincerely thank my advisor, Professor Bin Nie. Throughout the entire research journey, Professor Nie's wealth of knowledge and professional insights have provided me with invaluable guidance. Second, I wish to extend my gratitude to my colleagues in the laboratory. Your assistance and suggestions during the writing process have contributed to making this article's description more accurate and clear, adding substantial value to my research work. Lastly, I want to offer a special thanks to my family and friends. Your encouragement and support serve as the driving force behind my progress. At this pivotal moment, I would like to express my sincerest gratitude to all those who have lent their assistance to my research work. Without your support, I would not have been able to complete this study. Once again, my heartfelt thanks to each and every one of you!

### Funding

This research was supported by the National Natural Science Foundation of China (No. 82260849); the National Natural Science Foundation of China (No. 82260988); the National Natural Science Foundation of China (No. 61562045); and the Jiangxi University of Chinese Medicine Science and Technology Innovation Team Development Program (No. CXTD22015). The funders had no role in study design, data collection and analysis, decision to publish, or preparation of the manuscript.

### Grant Disclosures

The following grant information was disclosed by the authors:
National Natural Science Foundation of China: 82260849.
National Natural Science Foundation of China: 82260988.
National Natural Science Foundation of China: 61562045.
Jiangxi University of Chinese Medicine Science and Technology Innovation Team Development Program: CXTD22015.

### Competing Interests

The authors declare that they have no competing interests.

## Author Contributions

- Yuchao Zhang conceived and designed the experiments, performed the experiments, analyzed the data, performed the computation work, prepared figures and/or tables, authored or reviewed drafts of the article, and approved the final draft.
- Bin Nie conceived and designed the experiments, authored or reviewed drafts of the article, and approved the final draft.
- Jianqiang Du conceived and designed the experiments, authored or reviewed drafts of the article, and approved the final draft.
- Jiandong Chen conceived and designed the experiments, authored or reviewed drafts of the article, and approved the final draft.
- Yuwen Du performed the experiments, authored or reviewed drafts of the article, and approved the final draft.
- Haike Jin performed the experiments, analyzed the data, authored or reviewed drafts of the article, and approved the final draft.
- Xuepeng Zheng performed the experiments, authored or reviewed drafts of the article, and approved the final draft.
- Xingxin Chen analyzed the data, authored or reviewed drafts of the article, and approved the final draft.
- Zhen Miao analyzed the data, authored or reviewed drafts of the article, and approved the final draft.

## Data Availability

The data used in the experiment from The UCI Machine Learning Repository is available at https://archive.ics.uci.edu.

The code is available in the Supplemental Files.

## Supplemental Information

Supplemental information for this article can be found online at http://dx.doi.org/10.7717/peerj-cs.1711#supplemental-information.

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
