# Peer review of "Feature selection based on neighborhood rough sets and Gini index"

_PeerJ Computer Science, doi:10.7717/peerj-cs.1711_

## Round 0.1 · original submission · Minor Revisions

Hi,

you are requested to respond to each and every comment of the expert review.

/Khursheed

Reviewer 1 ·

Basic reporting

The article is written in clear English and the text is technically correct.
The article includes sufficient literature.
The structure of the article conforms to the acceptable format.
The submission is ‘self-contained,’ and includes relevant results.
Formal results include clear definitions of all terms.

Experimental design

The research is within Aims and Scope of the journal.
The submission clearly defines the research question.
The investigation has been conducted rigorously and to a high technical standard.
Methods are described with sufficient information.

Validity of the findings

The novelty of the recearch is stated.
The data on which the conclusions are based are provided. The data are robust and statistically sound.
The conclusions are a appropriately stated.

Additional comments

In Abstract and Conclusions you should give more detail about the obtained results.
You should specify the accuracy and mention practical application of you recearch In Abstract and Conclusions.
In Conclusions you should state the advantages of you approach in a comparison to the existing approaches.
What are the advantages of the proposed algorithms?
State clearly the novelty of your research in Conclusions.

Cite this review as

·

Basic reporting

.

Experimental design

.

Validity of the findings

.

Additional comments

The article is well written; however, it requires some minor revisions.
The figures and tables in the manuscript have been checked.

Revisions
Line 9-10, on page 3, please cite a couple recent studies which have used the technique.
Line 10, pg 3, did the authors mean “equivalent classes”, instead of “equivalence”.
Rephrase Line 27, pg 17. It is an important point, so rephrase such that it is easier to understand.
While only SVC and KNN algorithms have been tried in the experimental analysis section, I am wondering if it is possible for the authors to try a DLNN algorithm and XGBoost/CatBoost perhaps. That will strengthen the study in my opinion. XGBoost’s feature importance technique could also be used to provide comparison with the proposed method.
Figure 1 X axes in both the figures need to be renamed.
Correct the figure 2 caption please.
Figure 3 caption needs correction as well. Figure 5 X axis needs to be relabeled with full text.
Suggestion to convert Box 1 Algorithm 1 to a figure with a flowchart.
Overall the study is strong, but a bit more comparison to the existing feature importance techniques will make it even more strong.

---

## Round 0.2 · accepted · Accept

Thank you for addressing the reviewers' comments.